# A Retrospective Cohort Study of a Newly Proposed Criteria for Sporadic Creutzfeldt–Jakob Disease

**DOI:** 10.3390/diagnostics14212424

**Published:** 2024-10-30

**Authors:** Toshiaki Nonaka, Ryusuke Ae, Koki Kosami, Hiroya Tange, Miho Kaneko, Takehiro Nakagaki, Tsuyoshi Hamaguchi, Nobuo Sanjo, Yoshikazu Nakamura, Tetsuyuki Kitamoto, Yoshiyuki Kuroiwa, Kensaku Kasuga, Manabu Doyu, Fumiaki Tanaka, Koji Abe, Shigeo Murayama, Ichiro Yabe, Hideki Mochizuki, Takuya Matsushita, Hiroyuki Murai, Masashi Aoki, Koji Fujita, Masafumi Harada, Masaki Takao, Tadashi Tsukamoto, Yasushi Iwasaki, Masahito Yamada, Hidehiro Mizusawa, Katsuya Satoh, Noriyuki Nishida

**Affiliations:** 1Department of Molecular Microbiology and Immunology, Nagasaki University Graduate School of Biomedical Sciences, Nagasaki 852-8523, Japan; 2Japanese Prion Disease Surveillance Committee, Tokyo 187-8551, Japan; 3Division of Public Health, Center for Community Medicine, Jichi Medical University, Shimotsuke 329-0498, Japan; 4Department of Neurology, Kanazawa Medical University, Kahoku-gun 920-0293, Japan; 5Department of Internal Medicine, Division of Neurology, Kudanzaka Hospital, Tokyo 102-0074, Japan; 6Department of Neurology and Neurological Science, Tokyo Medical and Dental University Graduate School of Medical and Dental Sciences, Tokyo 113-8519, Japan; 7Department of Neurological Science, Tohoku University Graduate School of Medicine, Sendai 980-8574, Japan; 8Department of Neurology and Stroke Center, Teikyo University School of Medicine, Mizonokuchi Hospital, Kawasaki 213-8507, Japan; 9Department of Molecular Genetics, Brain Research Institute, Niigata University, Niigata 951-8585, Japan; 10Department of Neurology, Aichi Medical University, Nagakute 480-1195, Japan; 11Department of Neurology and Stroke Medicine, Yokohama City University Graduate School of Medicine, Yokohama 236-0004, Japan; 12National Center of Neurology and Psychiatry (NCNP), Tokyo 187-8551, Japan; 13Department of Neurology, Graduate School of Medicine, Dentistry and Pharmaceutical Science, Okayama University, Okayama 700-8558, Japan; 14Brain Bank for Neurodevelopmental, Neurological and Psychiatric Disorders, Molecular Research Center for Children’s Mental Development, United Graduate School of Child Development, Osaka University, Osaka 565-0871, Japan; 15Brain Bank for Aging Research, Tokyo Metropolitan Geriatric Hospital and Institute of Gerontology, Tokyo 173-0015, Japan; 16Department of Neurology, Faculty of Medicine and Graduate School of Medicine, Hokkaido University, Sapporo 060-8638, Japan; 17Department of Neurology, Osaka University Graduate School of Medicine, Osaka 565-0871, Japan; 18Department of Neurology, Kochi Medical School, Kochi University, Nankoku 783-8505, Japan; 19Department of Neurology, International University of Health and Welfare, Narita 286-8686, Japan; 20Department of Neurology, Tohoku University Graduate School of Medicine, Sendai 980-8574, Japan; 21Department of Neurology, Tokushima University Graduate School of Biomedical Sciences, Tokushima 770-8503, Japan; 22Department of Radiology, Tokushima University Graduate School, Tokushima 770-8503, Japan; 23Department of Clinical Laboratory, National Centre of Neurology and Psychiatry (NCNP), Tokyo 187-8551, Japan; 24Department of Neurology, National Center of Neurology and Psychiatry (NCNP), Tokyo 187-8551, Japan; 25Department of Neuropathology, Institute for Medical Science of Aging, Aichi Medical University, Nagakute 480-1195, Japan; 26Unit of Medical and Dental Sciences, Department of Health Sciences, Nagasaki University Graduate School of Biomedical Sciences, Nagasaki 852-8501, Japan; 27Department of Brain Research Unit, Leading Medical Research Core Unit, Nagasaki University Graduate School of Biomedical Sciences, Nagasaki 852-8501, Japan

**Keywords:** sporadic Creutzfeldt–Jakob disease, prion disease, biomarker, magnetic resonance imaging, real-time quaking-induced conversion, diagnostic criteria

## Abstract

Background/Objectives: Sporadic Creutzfeldt–Jakob disease (sCJD) is a fatal neurodegenerative disorder traditionally diagnosed based on the World Health Organization (WHO) criteria in 1998. Recently, Hermann et al. proposed updated diagnostic criteria incorporating advanced biomarkers to enhance early detection of sCJD. This study aimed to evaluate the sensitivity and specificity of Hermann’s criteria compared with those of the WHO criteria in a large cohort of patients suspected of prion disease in Japan. Methods: In this retrospective cohort study, we examined the new criteria using data of 2004 patients with suspected prion disease registered with the Japanese Prion Disease Surveillance (JPDS) between January 2009 and May 2023. Patients with genetic or acquired prion diseases or incomplete data necessary for the diagnostic criteria were excluded, resulting in 786 eligible cases. The sensitivity and specificity of the WHO and Hermann’s criteria were calculated by comparing diagnoses with those made by the JPDS Committee. Results: Of the 786 included cases, Hermann’s criteria helped identify 572 probable cases compared with 448 by the WHO criteria. The sensitivity and specificity of the WHO criteria were 96.4% and 96.6%, respectively. Hermann’s criteria demonstrated a sensitivity of 99.3% and a specificity of 95.2%, indicating higher sensitivity but slightly lower specificity. Fifty-five cases were classified as “definite” by both criteria. Conclusions: The findings suggest that Hermann’s criteria could offer improved sensitivity for detecting sCJD, potentially reducing diagnostic oversight. However, caution is advised in clinical practice to avoid misdiagnosis, particularly in treatable neurological diseases, by ensuring thorough exclusion of other potential conditions.

## 1. Introduction

Human prion diseases are untreatable and fatal neurodegenerative disorders characterized by the accumulation of misfolded prion proteins, known as PrP^Sc^ (scrapie prion protein). PrP^Sc^ aggregates and forms amyloid fibers, which deposit in the brain [1,2]. These diseases are broadly classified into sporadic, genetic, and acquired prion diseases. Sporadic Creutzfeldt–Jakob disease (sCJD) is not associated with mutations in the prion protein (*PRNP*) gene or a history of prion infection, such as those associated with dura mater or corneal transplants. The symptoms of sCJD include cognitive impairment, myoclonus, ataxia, weakness, and visual disturbances, with prognosis ranging from several weeks to a few months. Neuropathologically, sCJD is marked by spongiform changes and brain deposits of abnormal prion proteins.

Cases of sCJD are classified as definite, probable, or possible according to the World Health Organization (WHO) criteria (1998) (Appendix A) [3]. A definite diagnosis requires the detection of PrP^Sc^ in the central nervous system, which can be confirmed by immunocytochemistry and/or Western blotting to identify protease-resistant PrP and/or prion-specific fibrils. A probable diagnosis is based on the presence of several symptoms, including rapidly progressive dementia, myoclonus, visual disturbances, cerebellar dysfunction, pyramidal signs, extrapyramidal signs, and akinetic mutism. In addition to these symptoms, a probable diagnosis also requires either periodic sharp wave complexes (PSWCs) on electroencephalography (EEG) or positive results for cerebrospinal fluid (CSF) 14-3-3 protein. The detection of CSF 14-3-3 protein enhances the sensitivity and specificity for diagnosing sCJD compared with relying on PSWCs on EEG [4].

The MM1 type of sCJD, which accounts for approximately 70% of sCJD cases [5], progresses rapidly, often on a weekly basis. As a result, the WHO criteria rarely capture the full range of typical symptoms (I + two of II, as detailed in Appendix A) in the early stages. By the time these criteria are met, the disease has usually significantly advanced. These findings indicate the critical need for biomarkers that can enable establishing diagnosis at an earlier stage of the disease.

Recently, significant progress has been made in developing biomarkers for prion diseases, highlighting their importance for diagnostic testing. Brain magnetic resonance imaging (MRI) has become a crucial biomarker for diagnosing sCJD, particularly in its early stages. Diffusion-weighted imaging (DWI), a type of MRI scan, is particularly valuable as it can detect abnormalities earlier with greater sensitivity and specificity compared with PSWCs [6]. The real-time quaking-induced conversion (RT-QuIC) assay is another advanced method for detecting PrP^Sc^, even in minute amounts, through PrP^Sc^ amplification using recombinant prion protein [7,8]. The RT-QuIC assay allows for differentiation of patients with sCJD with high sensitivity and specificity [9]. Due to the development of biomarkers, Hermann et al. [10] incorporated these superior biomarkers into the WHO criteria, proposing new diagnostic criteria to enable earlier clinical diagnosis.

To determine whether Hermann’s criteria have improved the process of sCJD diagnosis of sCJD, we conducted a retrospective evaluation using data of patients registered with the Japanese Prion Disease Surveillance (JPDS) Committee. This study aimed to assess the utility and potential aspects of Hermann’s criteria by calculating their sensitivity and specificity.

## 2. Materials and Methods

### 2.1. Patients and Case Definition

Patients suspected of having prion disease were registered with the JPDS Committee (JPDSC). The surveillance committee determined diagnoses based on the clinical course, available data, and discussions with surveillance members for each case. This process involved collaboration with prion disease experts and incorporated epidemiology, neuroimaging, genetic analysis, CSF testing, Western blotting of the brain section in front region, and neuropathology. The diagnoses made were designated as JPDSC diagnoses. In this study, sCJD cases were classified as ‘definite’, ‘probable’, or ‘possible’ according to the JPDSC diagnoses. Cases diagnosed with other diseases or where sCJD was ruled out were classified as ‘non-prion disease’. Cases were classified as ‘unknown’ if they involved undetermined prion diseases or lacked sufficient symptoms for a definitive diagnosis, including cases of biomarkers suggesting prion disease where sCJD could not be ruled out.

### 2.2. Study Design

We retrospectively analyzed surveillance data from 2004 patients registered with the JPDS Committee between January 2009 and May 2023. These patients were either suspected of having prion diseases or exhibited progressive dementia requiring evaluation for prion diseases. Patients with genetic or acquired prion diseases were excluded. The inclusion criteria required *PRNP* gene sequence data without mutations and results from EEG, CSF assays, and MRI. The study period began in 2009 because 14-3-3 protein analysis was standardized that year, becoming a crucial diagnostic test for the JPDS Committee. This study used anonymized surveillance data.

Patients meeting the inclusion criteria were classified according to the WHO and Hermann’s diagnostic criteria using surveillance data, including their symptoms and biomarkers. The symptoms and signs identified based on the surveillance data were narrowed down to rapidly progressive dementia, progressive neuropsychiatric syndrome, myoclonus, visual disturbance, cerebellar disturbance, pyramidal signs, extrapyramidal signs, and akinetic mutism. The biomarkers extracted included PSWCs on EEG, elevated 14-3-3 protein in CSF, brain MRI, and RT-QuIC assay results. MRI data were limited to the presence of high intensity on DWI for the thalamus, basal ganglia, and cortical regions. Disease duration was calculated by subtracting the time of onset from the time of death and expressed in days. If the time of death was unknown, disease duration was considered to be more than two years. A diagnostic flowchart was created to apply the WHO and Hermann’s criteria (Figure 1). Patient consent was obtained in accordance with the Declaration of Helsinki. This study was approved by the Institutional Ethics Committee of Nagasaki University Graduate School of Biomedical Sciences (reference number 23042804-2).

The surveillance data of 2004 patients registered with the Japanese Prion Disease Surveillance Committee between January 2009 and May 2023 are analyzed. Patients with genetic or acquired prion diseases are excluded. The inclusion criteria are cases of available data on *PRNP* gene sequences, EEG, CSF assays, and MRI results. Diagnoses are established based on the WHO and Hermann’s criteria. Blue bold letters indicate classifications according to the WHO criteria, while yellow bold letters indicate classifications according to Hermann’s criteria.

### 2.3. Calculation of Sensitivity and Specificity

We considered the JPDSC diagnoses to be accurate clinically and collected cases of probable sCJD and non-prion diseases. We then applied both the WHO and Hermann’s criteria to these groups. Sensitivity was defined as the ratio of probable sCJD cases correctly identified by each criterion to the total number of probable cases according to the JPDSC diagnosis. Specificity was defined as the ratio of non-prion disease cases correctly identified by each criterion to the total number of non-prion disease cases according to the JPDSC diagnosis. False-positive cases were those diagnosed as non-prion diseases by the JPDSC but classified as probable sCJD cases by either the WHO or Hermann’s criteria. False-negative cases were those diagnosed as probable sCJD by the JPDSC but not meeting the criteria for either the WHO or Hermann’s.

### 2.4. Data on Biomarkers and Screening of PRNP Gene

Data of patients, including their biomarkers and *PRNP* gene screening, were entered into the JPDS. EEGs were conducted at each hospital, with the presence of PSWCs confirmed by the JPDS Committee, including a specialist in EEG for prion diseases. The Western blotting assay for 14-3-3 protein in the CSF was performed as previously described [11]. The RT-QuIC assay for CSF, known as the first-generation RT-QuIC assay, was conducted as previously described [7]. Brain MRI was performed at each hospital; three neuroradiologists, along with the JPDS Committee members, confirmed presence of hyperintensities on DWI in the thalamus, basal ganglia, cortex, and other sites. Cases of restricted diffusion in at least one lesion of the basal ganglia or cortex were classified as DWI positive in this data-based diagnosis. Genotype and mutations in the open reading frame of the *PRNP* gene were determined by sequence analysis as previously described [12].

### 2.5. Statistical Analysis

Statistical analysis was conducted using the R software environment for statistical computing (version 4.4.0, The R Foundation, Vienna, Austria). The Kruskal–Wallis test was performed, followed by Dunn’s test for post hoc analysis. Using Dunn’s test, the *p*-values from the Kruskal–Wallis test were corrected using the Bonferroni method. A significance level of 5% was set for all analyses.

## 3. Results

### 3.1. Diagnosis by the Japanese Prion Disease Surveillance Committee

We reviewed data of patients enrolled in surveillance between January 2009 and May 2023 to determine whether they had contracted prion diseases and, if so, the specific type of prion disease. Of the 2004 patients initially considered, we excluded 392 cases of genetic prion disease, 13 cases of iatrogenic prion disease, and none of the cases of variant prion disease, including suspected cases. Finally, a total of 786 cases met the inclusion criteria (Figure 1). The diagnoses of these 786 patients were analyzed and classified according to the JPDS Committee as follows: 55 definite cases, 419 probable cases, 118 possible cases, 48 unknown cases, and 146 non-prion disease cases (Table 1). These classifications were referred to as ‘JPDSC diagnoses’.

The unknown cases accounted for 6.1% of the 786 cases. Most patients were diagnosed with some type of disease. The Parchi’s classification is a method for categorizing sCJD. It is based on the molecular weight of the non-glycoform of PrP^Sc^, which defines type 1 and type 2, as well as the codon 129 polymorphism (MM, MV, and VV genotypes) [5]. This allows for classifying sCJD into six molecular subtypes. The main advantage of dividing CJD into six subtypes is that each subtype has different clinical manifestations, disease duration, and biomarker positive results. According to Parchi’s classification, the analysis of the 54 definite cases revealed the following distribution: 26 cases of MM1 type; 8 cases of MM1+2 type; 4 cases each of MM1+2C type and MM2 type; 3 cases each of MM2C type, MM2T type, and MV2 type; and 1 case each of MM2C+1 type, MV1+2 type, and MV2K type. Unfortunately, 1 of the 55 definite cases lacked data of the Western blotting assay and had an M/M type of codon 129 polymorphism. The categorization of non-prion disease cases is detailed in Appendix A.

### 3.2. Diagnosis According to WHO and Hermann’s Criteria

A total of 786 cases were evaluated using two separate sets of criteria. According to the WHO criteria, the classification included 55 definite cases, 448 probable cases, 14 possible cases, and 269 cases that did not meet the criteria. In contrast, applying Hermann’s criteria resulted in 55 definite cases, 572 probable cases, 3 possible cases, and 156 cases that did not fit the criteria (Table 1). Notably, the use of Hermann’s criteria led to a significant increase of 124 patients in the probable category. A total of 80 of the 124 cases were RT-QuIC positive. Among the remaining 44 RT-QuIC negative cases, all 44 cases had positive DWI findings; 16 had elevated 14-3-3 protein, and none showed positivity for PSWCs. Among the 124 increased cases, many fell into the probable category due to RT-QuIC even if they were not of fulfilled symptoms. A total of 122 of the 124 cases showed DWI positive and the remaining 2 cases had elevated 14-3-3 protein. Most cases of increases had positive biomarker on DWI. A total of 15 of the 124 cases of increases showed positivity for PSWCs. Among the 15 cases, 12 showed elevated 14-3-3 protein, 15 were positive on DWI and RT-QuIC. The PSWCs positive cases were characterized by owing biomarkers suggestive of sCJD. RT-QuIC and DWI were mainly involved in the cases of increases using Hermann’s criteria.

### 3.3. The Sensitivity and Specificity of Hermann’s Criteria Were Comparable to the WHO Criteria

To evaluate the performance of the two criteria, their sensitivity and specificity were assessed. Two cohorts were created: one consisting of probable cases identified through JPDSC diagnosis and another comprising non-prion disease cases. Each cohort was then evaluated using both WHO and Hermann’s criteria (Table 2).

In the probable case group, 404 cases met the WHO criteria, while 416 cases met Hermann’s criteria. In the non-prion disease group, 141 cases were excluded by the WHO criteria, compared with 139 cases excluded by Hermann’s criteria. These results correspond to a sensitivity of 96.4% and a specificity of 96.6% for the WHO criteria, versus a notable 99.3% sensitivity and 95.2% specificity for Hermann’s criteria. Hermann’s criteria demonstrated higher sensitivity and comparable specificity to the WHO criteria.

### 3.4. The Period from Onset to Probable Diagnosis

The period from onset to probable diagnosis was calculated for 437 out of 448 cases using the WHO criteria and 555 out of 572 cases using Hermann’s criteria. Regarding the WHO criteria, the median period was 57 days (interquartile range [IQR], 35–104; range, 1–4628 days). For Hermann’s criteria, the median period was 59 days (IQR, 32.5–118; range, 1–4628 days). There was no significant difference between the two criteria (*p* = 1.00, Figure 2). In cases classified as probable only by Hermann’s criteria and not by the WHO criteria (118 out of 124 cases), the median period was 116 days (IQR, 53–213; range, 1–1460 days). This period was significantly longer compared with the WHO criteria (*p* = 0.000001, Figure 2). The cases meeting only Hermann’s criteria apparently had a longer duration from onset to probable diagnosis. Atypical cases, such as MM2 cortical type, progressed slowly and were challenging to diagnose using the WHO criteria. Consequently, these cases were identified only by Hermann’s criteria, suggesting that they took longer to reach the probable category under Hermann’s criteria.

The box-and-whisker plot illustrates the distribution of the period from onset to probable diagnosis for the WHO criteria, Hermann’s criteria, and cases classified as probable only by Hermann’s criteria. The term ‘only Hermann’s criteria’ refers to cases that are classified as probable only by Hermann’s criteria and not by the WHO criteria. The plot shows that there are more outliers in both the WHO and Hermann’s criteria. The vertical axis represents the period in days on a logarithmic scale.

### 3.5. Hermann’s Criteria Have a Specificity as High as the WHO Criteria After Exclusion Diagnosis

The false-positive cases included three cases identified using the WHO criteria and five cases identified using Hermann’s criteria (Table 2). If exclusion diagnoses had not been performed, there would have been 23 false-positive cases using Hermann’s criteria and 9 using the WHO criteria, as these cases would have been incorrectly classified as probable (Table 3). These cases were categorized as non-prion diseases in the JPDSC diagnoses but were unfortunately not pathologically confirmed. To determine if exclusion diagnoses could be accurately ruled out, we investigated several factors: symptom improvement, MRI findings, response to steroid therapy, brain disorder pathophysiology, and pathology, to assess if they could be pathologically excluded.

A total of 18 out of the 23 cases could be diagnosed by exclusion using Hermann’s criteria. Among these 18 cases, 14 showed DWI positivity, 11 had 14-3-3 protein positivity, and 4 exhibited PSWCs positivity. None of the 18 cases tested positive for RT-QuIC. Among these 18 cases, 6 were diagnosed with autoimmune encephalitis and 3 with epilepsy. Of the six cases of autoimmune encephalitis, five were positive on DWI and three for 14-3-3 protein, without cases showing PSWCs or RT-QuIC positivity. The three cases of epilepsy all exhibited positivity both on DWI and for 14-3-3 protein.

The five false-positive cases showed positivity for various biomarkers: three had DWI positivity, two had 14-3-3 protein positivity, two exhibited PSWCs positivity, and one was positive for RT-QuIC. These five cases included two of encephalopathy, one of non-convulsive epileptic status, one of frontotemporal dementia, and one of an alcohol-related disorder. Further details are presented in Appendix A.

### 3.6. The False-Negative Cases and Biomarkers

Among the clinically identified probable cases, 14 were classified as false negatives according to the WHO criteria, while only 3 were false negatives using Hermann’s criteria (Table 2). Notably, the three cases classified as false negatives by Hermann’s criteria were also false negatives by the WHO criteria. Among the 14 false negatives using the WHO criteria, 6 cases showed positivity for PSWCs, 12 had elevated 14-3-3 protein levels, all 14 had positive DWI findings, and 11 displayed RT-QuIC positivity and progressive neuropsychiatric syndromes.

When further categorized based on the presence of typical symptoms (I + two of II, as defined in Appendix A), 4 patients with typical symptoms and 10 without them were identified. Notably, the four patients with typical symptoms lacked PSWCs positivity, only two had elevated 14-3-3 protein levels, all four exhibited positive DWI findings, and three showed RT-QuIC positivity. The four cases of typical symptoms were classified as probable cases according to Hermann’s criteria but were not as probable cases according to the WHO criteria, resulting in false negatives. Among the 10 cases without typical symptoms, biomarker positivity was observed: 6 cases showed PSWCs positivity, all 10 had elevated 14-3-3 protein levels and positive DWI findings, and 8 exhibited RT-QuIC positivity and progressive neuropsychiatric syndromes. The 10 false-negative cases without typical symptoms were considered clinically probable cases by JPDSC.

Notably, one false-negative case using Hermann’s criteria presented with typical symptoms but tested negative for PSWCs, 14-3-3 protein, DWI, and RT-QuIC. In this case, diffusion restriction on DWI was isolated to the thalamus, which is considered an atypical DWI finding according to Hermann’s criteria. Despite the absence of typical symptoms, all 14 false-negative cases were clinically deemed probable due to their disease course and biomarker positivity.

## 4. Discussion

A retrospective cohort study of 786 patient records was conducted using two diagnostic criteria, revealing that Hermann’s criteria led to an increase of 124 probable cases, significantly enhancing the diagnostic accuracy for sCJD. This improvement is attributed to the incorporation of novel biomarkers, MRI abnormalities, and 14-3-3 protein in CSF, without the constraint of disease duration. Hermann’s criteria identified the same number of definite cases as the WHO criteria. The sensitivity and specificity of Hermann’s criteria were comparable to those of the WHO criteria when exclusion diagnoses were considered. However, without exclusion diagnoses, Hermann’s criteria enabled categorizing more non-prion disease cases as probable, resulting in a higher number of false-positive diagnoses. This increased likelihood of misclassifying non-prion disease cases was due to Hermann’s criteria incorporating highly sensitive biomarkers, such as DWI and 14-3-3 protein, and not considering the duration of illness for 14-3-3 protein.

Out of the 23 false positives in case of insufficient exclusion diagnosis, 73.9% were positive for DWI. Previous studies have shown that the sensitivity and specificity of DWI were 91–92.3% and 93.8–97%, respectively, for CJD [6,13,14]. The JPDS identified specific sites of restricted diffusion, including the cortex, basal ganglia, and thalamus, while Hermann’s criteria typically enabled identifying the caudate, caudate/putamen, caudate/putamen/thalamus, or at least two cortical regions such as the temporal, parietal, and occipital regions on brain MRI. Therefore, restricted diffusion in the cortical region, including the frontal cortex, was likely to be considered DWI positivity in this study. Cases of restricted diffusion designated by the JPDS were listed to have DWI positivity even if the restricted diffusion subsequently disappeared. These represent the limitations of this study. Furthermore, we evaluated the sensitivity and specificity of DWI based on previous studies. A meta-analysis highlighted a limitation: half of the included studies lacked control groups [13]. We reviewed diagnoses in patients with non-prion disease from eight studies with controls and two additional retrospective studies [6,14]. These studies included 552 cases of sCJD and 279 cases of non-prion disease. Among the controls in these studies, the prevalence of epilepsy, autoimmune encephalitis, and neurodegenerative diseases were 0.7%, 12.2%, and 35.5%, respectively. In contrast, in our study, these percentages were 15.4%, 13.4%, and 32.2%, respectively. The significant difference in epilepsy rates suggests that misdiagnoses based on MRI findings could occur. As a result, MRI findings at a single time point may sometimes be indicative of other conditions, such as epilepsy and encephalitis.

The diagnostic application of MRI modalities needs refinement. A study has reported that hypointensity of the apparent diffusion coefficient (ADC) on MRI is observed in subcortical DWI hyperintensities associated with sCJD but not in autoimmune encephalopathy [14]. To minimize false-positive cases, it is advisable to use ADC findings as a reference rather than as definitive evidence. Recently, MRI with arterial spin labelling (ASL) has been reported to demonstrate reduced regional cerebral blood flow in CJD patients [15]. To differentiate CJD from other conditions, such as epilepsy—particularly non-convulsive status epilepticus—MRI with ASL imaging can be highly useful. In epilepsy, MRI with ASL imaging typically shows increased regional cerebral blood flow [16,17]. Epilepsy often exhibits either ictal or peri-ictal hyperperfusion or postictal or interictal hypoperfusion on MRI with ASL perfusion [18]. Therefore, incorporating both DWI and ASL imaging into the diagnostic criteria for sCJD is anticipated to enhance specificity. The addition of ASL imaging improves specificity compared to relying on MRI findings alone.

Hermann’s criteria demonstrated specificity comparable to the WHO criteria when conducting exclusion diagnoses. However, meeting Hermann’s criteria does not guarantee that sCJD is ruled out. Differential diagnosis remains crucial. Key conditions that can mimic sCJD include immune-mediated encephalitis, brain infections, toxic or metabolic brain disorders, neoplastic or paraneoplastic conditions, and vascular disorders [19]. Immune-mediated encephalitis is an autoimmune disorder of the central nervous system that responds to immunotherapy. This category includes N-methyl-D-aspartate receptor antibody encephalitis, voltage-gated potassium channel complex antibody encephalitis, and Hashimoto’s encephalitis. Brain infections encompass viral encephalitis, progressive multifocal leukoencephalopathy, and subacute sclerosing panencephalitis, often involving white matter lesions. Diagnosis of these conditions typically requires additional tests. Toxic and metabolic brain disorders can often be identified through initial laboratory tests assessing sodium, calcium, magnesium, glucose, and thyroid function. Specific conditions to consider include Wernicke’s encephalopathy, hepatic failure, and hyperammonemia. Neoplastic and paraneoplastic conditions, such as primary central nervous system lymphoma, carcinomatosis, and intravascular lymphoma, should also be evaluated. Vascular disorders include ischemic stroke, dural arteriovenous fistula, posterior reversible encephalopathy syndrome, and primary central nervous system vasculitis. MRI and contrast radiography are primary diagnostic tools for these conditions. Key considerations include evaluating the response to steroid therapy, examining for other neurodegenerative diseases, and reviewing the patient’s alcohol history. A case series of four steroid-responsive encephalopathy showed they had only Alzheimer’s Disease-related findings and no prion disease pathologically [20]. Our search of the literature did not reveal that none of cases with steroid-responsive encephalopathy were identified as prion disease pathologically. Therefore, suspected cases of CJD should be treated with steroids. Additionally, monitoring biomarkers, assessing changes in high signal intensity on DWI over time, and periodically reapplying Hermann’s criteria are important. If diagnoses remain uncertain, retesting biomarkers after a few weeks can be effective.

The RT-QuIC assay has shown a sensitivity of 80–82% and a specificity of 99–100% in previous studies [7,21], with an exceptional specificity of approximately 100%. However, due to its lower sensitivity, the RT-QuIC assay alone may result in false negatives. To minimize false-negative results, combining the RT-QuIC assay with other highly sensitive biomarkers is recommended. Considering alternative diagnoses of prion disease when a case is positive for both 14-3-3 protein and DWI but negative for the RT-QuIC assay is essential. Among 419 patients with probable sCJD, 40 had positive 14-3-3 protein results, positive DWI findings, and negative RT-QuIC assay results. This underscores the importance for including both the 14-3-3 protein and RT-QuIC assays in CSF testing. It is also noteworthy that while the 14-3-3 protein test is used worldwide, the RT-QuIC assay for CSF is available in only approximately 30 countries. Consequently, both the 14-3-3 protein and RT-QuIC assay have been incorporated into the diagnostic criteria for CSF tests as key biomarkers. This study utilized the first-generation RT-QuIC assay was employed because it was adopted by the JPDS Committee. We are currently reanalyzing CSF samples using the second-generation RT-QuIC assay. This decision was made due to the superior sensitivity of the second-generation RT-QuIC assay compared with the first-generation RT-QuIC [9,22,23]. Furthermore, new detection methods for other tissues or body fluids may be necessary to facilitate early diagnosis.

In Western countries and Japan, useful tools such as MRI, CSF biomarkers analysis, and RT-QuIC are widely available and assist in establishing diagnosis. However, in other regions, these tools are less accessible. Moreover, cultural factors might influence the diagnosis of sCJD. In recent years, the spread of MRI and CSF biomarkers has been improving this situation.

A 73-year-old man exhibited signs of rapidly progressive dementia, altered consciousness, and myoclonus. Initially suspected of having refractory non-convulsive status epilepticus. His symptoms included sudden onset of aphasia, disappearance of PSWCs during sleep, lack of response to steroid therapy, and persistent high-amplitude PSWCs despite disease progression. However, he tested positive using the RT-QuIC assay, had PSWCs on EEG, elevated CSF tau protein levels, and high-intensity findings on DWI in the cortex, caudate nucleus, and striatum. These findings suggested a possible prion carrier or prion disease. Neuropathological confirmation of prion disease is crucial in cases of refractory pathology.

This study has some limitations. First, the study period began after the standardization of CSF 14-3-3 protein in 2009, while the RT-QuIC assay was developed afterward. Therefore, it cannot be guaranteed that every case strictly met all the same criteria. Second, the database used in this study records the presence or absence of DWI, T2-weighted imaging, and fluid-attenuated inversion recovery, but the information on DWI hyperintensity is limited to only three brain sites: the cerebral cortex, thalamus, and basal ganglia. Therefore, whether the DWI hyperintensities were in at least two regions of the cerebral cortex and not including in the frontal cortex was not certain. Furthermore, assessing the presence of ADC hypointensity, as specified using Hermann’s criteria was impossible. Third, the small number of definite cases limited to calculate sensitivity and specificity accurately in this study. This limitation arose from the few patients who underwent autopsy in Japan. Although comparing the diagnostic accuracy of clinical criteria with neuropathological confirmation is necessary, regional limitations resulted in a scarcity of cases of pathological confirmation. Further research is needed to investigate both definite prion disease cases and non-prion disease cases using Hermann’s criteria. Despite these limitations, Hermann’s criteria, which incorporate new biomarkers, may be valuable for diagnosing sCJD in challenging pathological research conditions.

## 5. Conclusions

The incorporation of a novel biomarker has led to an increase in the number of probable sCJD cases. The reduced number of false-negative instances observed using Hermann’s criteria suggests that early-stage sCJD cases were identified through a positive RT-QuIC assay. However, the heightened sensitivity of Hermann’s criteria has also resulted in more false-positive cases related to treatable conditions. Therefore, exercising caution when using Hermann’s criteria is crucial for avoiding overlooking treatable conditions and over-diagnosing sCJD. Applying Hermann’s criteria to patients with rapidly progressive dementia without a thorough differential diagnosis may lead to a mistaken diagnosis of sCJD and potentially deprive them of appropriate treatment opportunities.

## Figures and Tables

**Figure 1 diagnostics-14-02424-f001:**
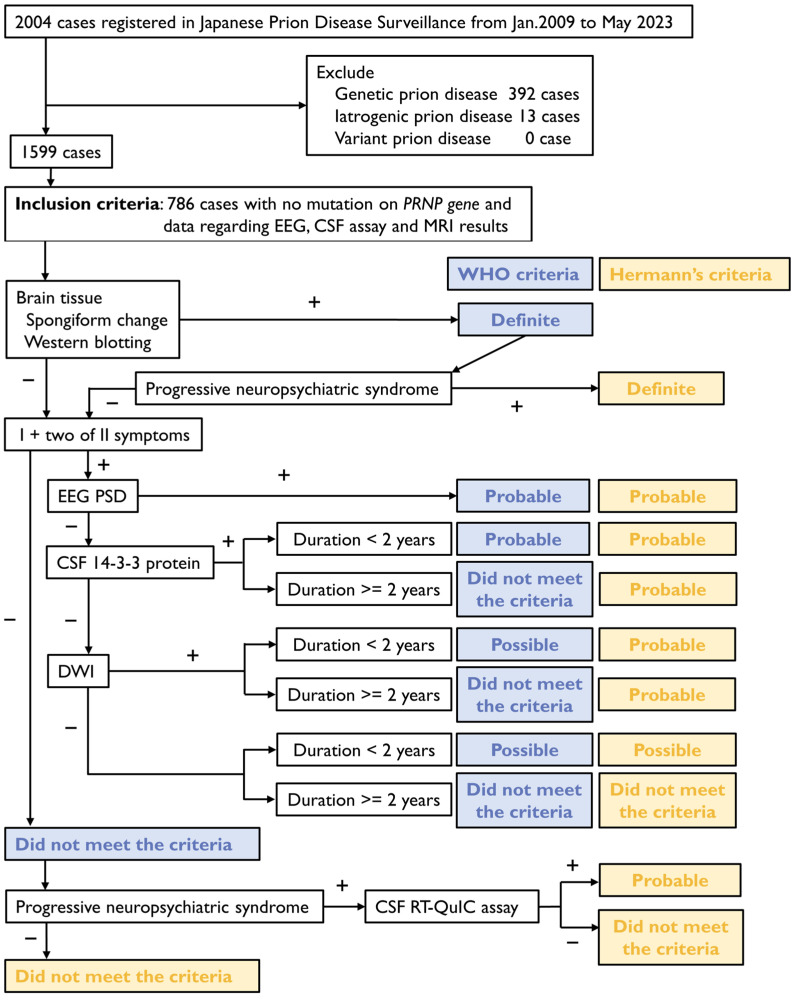
Diagnostic flowchart of the World Health Organization (WHO) and Hermann’s criteria for sporadic Creutzfeldt–Jakob disease (CJD). EEG, electroencephalography; CSF, cerebrospinal fluid; MRI, magnetic resonance imaging; DWI, diffusion-weighted imaging; RT-QuIC, real-time quaking-induced conversion.

**Figure 2 diagnostics-14-02424-f002:**
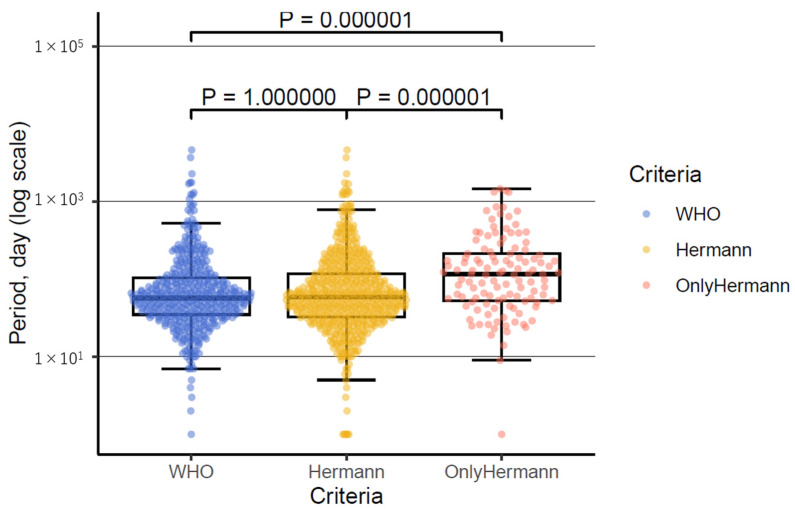
Distribution of period from onset to probable diagnosis using the WHO criteria, Hermann’s criteria, and only Hermann’s criteria.

**Table 1 diagnostics-14-02424-t001:** JPDSC diagnosis according to the WHO and Hermann’s criteria.

JPDSC Diagnosis		
JPDSC diagnosis		Number
Definite case		55
Probable case		419
Possible case		118
Unknown		48
Non-prion disease		146
Amount		786
The WHO and Hermann’s criteria		
Diagnosis	WHO	Hermann
Definite case	55	55
Probable case	448	572
Possible case	14	3
Case which did not meet the criteria	269	156
Amount	786	786

The upper left table displays the distribution of diagnoses according to the surveillance committee (JPDSC diagnosis). The lower right table shows the distribution of diagnoses based on WHO and Hermann’s criteria. The data include symptoms and biomarkers from the JPDS data, such as periodic sharp wave complexes, CSF assay results, and MRI findings. WHO, World Health Organization; CSF, cerebrospinal fluid; JPDS diagnosis, diagnosis by the Japanese Prion Disease Surveillance Committee.

**Table 2 diagnostics-14-02424-t002:** Two cohorts of probable case and non-prion disease case evaluated according to the WHO and Hermann’s criteria.

Probable Case Group (JPDSC Diagnosis)					419
WHO criteria	Probable case	404	Hermann’s criteria	Probable case	416
	Possible case	1		Possible case	0
	Case which did not meet the WHO criteria	14 ^a^		Case which did not meet Hermann’s criteria	3 ^a^
Non-prion disease group (JPDSC diagnosis)					146
WHO criteria	Probable case	3 ^b^	Hermann’s criteria	Probable case	5 ^b^
	Possible case	2		Possible case	2
	Case which did not meet the WHO criteria	141		Case which did not meet Hermann’s criteria	139

The WHO and Hermann’s criteria were applied for 419 probable cases and 146 non-prion disease cases clinically identified by the surveillance committee. ^a^ False-negative cases for each criterion. ^b^ False-positive cases for each criterion.

**Table 3 diagnostics-14-02424-t003:** The list of false-positive cases according to the WHO and Hermann’s criteria.

Case	Age/Sex	False Positive		Biomarkers				Exclusion Diagnosis					Exclusion Diagnosis	Clinical Diagnosis
		WHO	Hermann’s	DWI	PSWCs	14-3-3	QuIC	Improvement of Symptoms	Improvement of DWI	Response for Steroid Therapy	Pathophysiology of Brain Disorder	Pathology		
1	73/M	+	+	+	+	+	+	−	−	−	−	N.A.	Impossible	Non-convulsive status epilepticus
2	71/M	+	+	−	−	+	−	−	−	−	−	N.A.	Impossible	Encephalitis
3	74/F	+	+	−	+	−	−	−	−	−	−	N.A.	Impossible	Neurodegenerative disease (frontotemporal lobar degeneration),
														hypoxemia
4	50/M		+	+	−	−	−	−	−	−	−	N.A.	Impossible	Alcohol-related disorder (alcohol-related ataxia) or
														neurodegenerative disease (spinocerebellar degeneration) suspected
5	65/M		+	+	−	−	−	−	−	−	−	N.A.	Impossible	Encephalopathy
6	45/M	+ w/o	+ w/o	−	+	+	−	−	−	−	−	Ruled out	Possible	Non-prion disease, neuropathological
												pathologically		
7	70/F	+ w/o	+ w/o	+	+	+	−	−	+	−	−	N.A.	Possible	Status epilepticus, neurodegenerative disease (corticobasal degeneration)
8	85/F	+ w/o	+ w/o	−	−	+	−	−	−	−	+	N.A.	Possible	Multiple brain metastases from lung cancer, hydrocephalus
9	88/F	+ w/o	+ w/o	+	−	+	−	−	−	−	−	Ruled out	Possible	Brain infarction (basal ganglia), Parkinson’s disease
												pathologically		
10	52/F	+ w/o	+ w/o	+	+	−	−	+	−	−	−	N.A.	Possible	Alcohol-related disorder
11	45/M	+ w/o	+ w/o	+	+	−	−	+	−	−	−	N.A.	Possible	Alcohol-related disorder (alcohol-related encephalopathy)
12	81/F		+ w/o	−	−	+	−	+	−	+	−	N.A.	Possible	Autoimmune encephalitis (anti-NMDA receptor encephalitis)
13	62/F		+ w/o	−	−	+	−	−	−	−	+	N.A.	Possible	Progressive multifocal leukoencephalopathy,
														systemic lupus erythematosus
14	46/M		+ w/o	+	−	+	−	−	−	−	+	N.A.	Possible	Postconvulsive encephalopathy, folic acid deficiency
15	70/M		+ w/o	+	−	+	−	−	+	−	−	N.A.	Possible	Drug-induced parkinsonism,
														epilepsy (CSE, post convulsive status encephalopathy), hypoxemia
16	71/F		+ w/o	+	−	+	−	+	−	+	−	N.A.	Possible	Autoimmune encephalitis
17	39/M		+ w/o	+	−	+	−	−	+	−	−	N.A.	Possible	Postconvulsive encephalopathy
18	51/F		+ w/o	+	−	+	−	+	−	+	−	N.A.	Possible	Autoimmune encephalitis
19	75/M		+ w/o	+	−	−	−	−	−	−	+	N.A.	Possible	Paraneoplastic syndrome
20	65/M		+ w/o	+	−	−	−	+	−	+	−	N.A.	Possible	Hashimoto’s encephalopathy
21	83/M		+ w/o	+	−	−	−	−	−	+	−	N.A.	Possible	Autoimmune encephalitis
22	30/F		+ w/o	+	−	−	−	+	−	+	−	N.A.	Possible	Autoimmune encephalitis (anti-NMDA receptor encephalitis)
23	72/M		+ w/o	+	−	−	−	+	−	−	−	N.A.	Possible	Dementia due to other causes

After ruling out other diseases based on several criteria, including symptom improvement, DWI findings, response to steroid therapy, the pathophysiology of brain disorders, and pathological examination, five cases were identified as false positives according to Hermann’s criteria. A total of 23 cases classified as probable according to Hermann’s criteria in the absence of an exclusion diagnosis. + w/o: + without exclusion diagnosis, N.A.: not autopsied. DWI: diffusion-weighted imaging, PSWCs: periodic sharp wave complexes, 14-3-3: 14-3-3 protein in CSF, QuIC: real-time quaking-induced conversion assay in CSF, NMDA: N-methyl-D-aspartic acid, and CSE: convulsive status epilepticus.

## Data Availability

The raw data supporting the conclusions of this article cannot be shared publicly due to the participant privacy. The data will be shared on reasonable request to the corresponding author.

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
