# Peer review of "A Retrospective Cohort Study of a Newly Proposed Criteria for Sporadic Creutzfeldt–Jakob Disease"

_diagnostics, 2024, doi:10.3390/diagnostics14212424_

Round 1
Reviewer 1 Report
Comments and Suggestions for Authors
The paper by Toshiaki Nonaka et al. is a retrospective study evaluating the implications, in terms of diagnostic capacity on sporadic CJD, of the introduction of newer biomarkers (neurophysiological, neuroimaging, and laboratory) versus older criteria in a large retrospective cohort of patients. The study is interesting, its methodology appears appropriate and provides useful suggestions for clinical practice and for future research on the topic. I have only a few suggestions for the Authors:
- The Authors correctly cite Parchi's classification but without elaborating on it. Some more information might be useful to provide the non-expert reader with a key and help understand the implications;
- If possible, I suggest delving into the abnormalities found on neuroimaging; for example, are there differences, in terms of diagnostic accuracy, between cortical ribboning and hyperintensity of specific structures (e.g., the pulvinar, but not only)?
- In my opinion, it would be important to place more value on clinical presentation; are there initial elements that can more correctly direct the diagnosis (at least compared to more specific ones)?
- The EEG is only evaluated in terms of the most characteristic abnormalities with respect to this pathological condition, while there is no in-depth discussion as to whether some of the patients had also had seizures patterns; this could be useful, and the implications should be discussed;
- The work does not differentiate some rare variants of sCJD, such as Oppenheimer-Brownell and, above all, Heidenhain. This would be particularly relevant, as these are some of those most easily interpreted as different clinical pictures, especially at the onset. For example, some interesting evidence in the literature reports cases of Heidenhain variant simulating PRES. I suggest citing these references and discussing their implications in the Discussion to further the differential diagnosis of these complex conditions;
- Did any patients also have data on very recent methods, e.g. RT-QuiC for Tau protein?
- Are studies similar to the one presented by the Authors available in different countries, e.g., in the Western world? Discussing them could be useful not only to evaluate differences in the application of diagnostic criteria in other healthcare settings but also to assess whether cultural factors may modify the diagnostic delay.
Author Response
Our point-by-point response to the comments by Reviewer 1 is given below:
Thank you very much for taking the time to review this manuscript. Please find the detailed responses below and the corresponding revisions/corrections highlighted/in track changes in the re-submitted files.
Comments 1: The Authors correctly cite Parchi's classification but without elaborating on it. Some more information might be useful to provide the non-expert reader with a key and help understand the implications;
Response 1: Thank you for this suggestion. We agree with you. Therefore, we have added a highlighted text to lines 212-217 of page 5 to elaborate on Parchi's classification.
Comments 2: If possible, I suggest delving into the abnormalities found on neuroimaging; for example, are there differences, in terms of diagnostic accuracy, between cortical ribboning and hyperintensity of specific structures (e.g., the pulvinar, but not only)?
Response 2: Thank you very much for your meaningful questions and feedback. Disease differentiation requires not only DWI but also FLAIR imaging. Our study focused only on DWI. In particular, we examine all cases for the presence or absence of high signal on DWI, which is currently not possible in this study, as you pointed out. We apologize for this.
Comments 3: In my opinion, it would be important to place more value on clinical presentation; are there initial elements that can more correctly direct the diagnosis (at least compared to more specific ones)?
Response 3: Thank you for your excellent query. As you pointed out, early diagnosis is very important. However, this study was retrospective in nature and focused on diagnosis at the time of symptom onset, which makes it difficult to study early diagnosis. Nevertheless, we agree that early diagnosis is very important and will consider it a topic in future research. In terms of early diagnosis, diagnosing the disease when there is only one clinical symptom or no symptoms is important; the Hermann criteria require at least two symptoms. However, early diagnosis requires diagnosis when these symptoms are not present. Further case ascertainment may be required. Moreover, based on data from a previous Japanese surveillance committee, the initial symptoms vary. Additionally, we would like to note that thoroughly verifying all data would take over a year. We appreciate your understanding regarding the time required for such comprehensive validation.
Comments 4: The EEG is only evaluated in terms of the most characteristic abnormalities with respect to this pathological condition, while there is no in-depth discussion as to whether some of the patients had also had seizures patterns; this could be useful, and the implications should be discussed;
Response 4: It is known that electroencephalography (EEG) tests of sCJD show slow waves before the appearance of PSWCs. We apologize that the EEG data available in the surveillance included only slow waves and PSWCs and did not comprise data on seizure patterns.
Comments 5: The work does not differentiate some rare variants of sCJD, such as Oppenheimer-Brownell and, above all, Heidenhain. This would be particularly relevant, as these are some of those most easily interpreted as different clinical pictures, especially at the onset. For example, some interesting evidence in the literature reports cases of Heidenhain variant simulating PRES. I suggest citing these references and discussing their implications in the Discussion to further the differential diagnosis of these complex conditions;
Response 5: Thank you for your insightful suggestions. Of the 786 cases that met the inclusion criteria, only one case was Heidenhain variant of CJD. Indeed, Oppenheimer-Brownell and Heidenhain variants often manifest cerebellar and visual symptoms first, making them easily mistaken for other neurological disorders, which requires clinical caution. We believe that including these rare forms of CJD in the differential diagnosis is important. However, this time we focused on making diagnosing by adding biomarkers. Therefore, it may be considered an overdiscussion. We will consider this as a future task.
Comments 6: Did any patients also have data on very recent methods, e.g. RT-QuiC for Tau protein?
Response 6: Thank you for your question. We apologize that data on very recent methods, e.g. RT-QuiC for Tau protein was not included in the surveillance data within terms shown in our manuscript.
Comments 7: Are studies similar to the one presented by the Authors available in different countries, e.g., in the Western world? Discussing them could be useful not only to evaluate differences in the application of diagnostic criteria in other healthcare settings but also to assess whether cultural factors may modify the diagnostic delay.
Response 7: Thank you for pointing this out. We agree with you regarding this comment. The amended diagnostic criteria are available in the Western world. Therefore, we added the below part to lines 471-475 of page 13.
In the Western countries and Japan, useful tools such as MRI, CSF biomarkers, and RT-QuIC are widely available and assist in establishing diagnosis. However, in other regions, these tools are less accessible. Moreover, cultural factors might influence the diagnosis of sCJD. In recent years, the spread of MRI and CSF biomarkers has been improving this situation.
Reviewer 2 Report
Comments and Suggestions for Authors
In this manuscript, the authors retrospectively evaluate the use of the novel Hermann-criteria for diagnosis of sporadic CJD and compare the outcome to the use of the previous WHO criteria. They show that in their retrospective patient data the new Hermann criteria had an increased sensitivity for the detection of sCJD, but that exclusion diagnosis needs to be performed in order to increase the specificity to that of the WHO criteria. This study is of interest to many researchers in the field of prion diseases.
The approach of the study is well designed and appropriately explained to the reader. However, a few aspects could be improved:
1) The diagnosis listed in the JPDS was considered accurate. It has not become clear to me why this classification was used as "accurate" because the majority of cases was classified "probable" or "possible". It is a bit disturbing to use these assignments as a kind of gold standard. The authors should better explain and give the reasons for using the JPDS diagnosis as "accurate" diagnosis.
2) As I understand, the sensitivity of the criteria was assessed by measuring the percentage of cases which were classified as "probable" by JPDS and were also classified as "probable" by the WHO and Hermann criteria. This approach overlooks the 124 additional cases classified as "probable" by the Hermann criteria but were classified as "possible" by JPDS. This is not really reflected in the reported increase of sensitivity from 96.4 to 99.3%, as this difference only applies to the 419 cases rated as "probable" by JPDS. The finding, that by using the Hermann criteria the number of cases which could be newly be classified as "probable" could be increased by approx. 30% is not well reflected in the presentation of the results, although in my opinion this might be the most important aspect of this study.
2) The special role of the RT-QuIC should be discussed more extensively. It is this novel test which constitutes the major difference between the WHO and Hermann criteria. The major problems arising from the application of the Hermann criteria are due to the low sensitivity of around 80% of this test. As it seems that this problem could be much improved with the 2nd generation RT-QuIC reaching sensitivity of near 100% while maintaining near 100% specificity, this would change the whole diagnostic field. The future of sCJD diagnosis under this aspect should be discussed more extensively.
Round 2
Reviewer 1 Report
Comments and Suggestions for Authors
I thank the Authors for their work, which improved the quality of their paper. No further comment